# Fibrinogen Glycation and Presence of Glucose Impair Fibrin Polymerization—An In Vitro Study of Isolated Fibrinogen and Plasma from Patients with Diabetes Mellitus

**DOI:** 10.3390/biom10060877

**Published:** 2020-06-07

**Authors:** Boguslawa Luzak, Magdalena Boncler, Marcin Kosmalski, Ewelina Mnich, Lidia Stanczyk, Tomasz Przygodzki, Cezary Watala

**Affiliations:** 1Department of Hemostasis and Hemostatic Disorders, Chair of Biomedical Sciences, Medical University of Lodz, Mazowiecka 6/8, 92-216 Lodz, Poland; magdalena.boncler@umed.lodz.pl (M.B.); mnich.ewelina@gmail.com (E.M.); lidia.stanczyk@umed.lodz.pl (L.S.); tomasz.przygodzki@umed.lodz.pl (T.P.); cezary.watala@umed.lodz.pl (C.W.); 2Department of Clinical Pharmacology, 1st Chair of Internal Medicine, Medical University of Lodz, Kopcinskiego 22, 90-153 Lodz, Poland; marcin.kosmalski@umed.lodz.pl

**Keywords:** fibrin polymerization, hyperglycemia, glycated fibrinogen, diabetes mellitus

## Abstract

Background: Fibrin formation and structure may be affected by a plethora of factors, including both genetic and posttranslational modifications, such as glycation, nitration or acetylation. Methods: The present study examines the effect of fibrinogen glycation on fibrin polymerization, measured in fibrinogen concentration-standardized plasma of subjects with type 2 diabetes mellitus (T2DM) and in a solution of human fibrinogen exposed to 30 mM glucose for four days. Results: The fibrin polymerization velocity (V_max_) observed in the T2DM plasma (median 0.0056; IQR 0.0049‒0.0061 AU/s) was significantly lower than in non-diabetic plasma (median 0.0063; IQR 0.0058‒0.0071 AU/s) (*p* < 0.05). Furthermore, significantly lower V_max_ was observed for glucose-treated fibrinogen (V_max_ 0.046; IQR 0.022‒0.085 AU/s) compared to control protein incubated with a pure vehicle (V_max_ 0.053; IQR 0.034‒0.097 AU/s) (*p* < 0.05). The same tendency was observed in the fibrinogen samples supplemented with 6 mM glucose just before measurements. It is assumed that glucose may affect the ability of fibrinogen to form a stable clot in T2DM subjects, and that this impairment is likely to influence the outcomes of some diagnostic assays. As the example, the impaired clotting ability of glycated fibrinogen may considerably influence the results of the standard Clauss method, routinely used to determine fibrinogen concentration in plasma. The stoichiometric analysis demonstrated that spontaneous glycation at both the sites with high and low glycation potential clearly dominated in T2DM individuals in all fibrinogen chains.

## 1. Introduction

Fibrinogen (340 kDa) is a large glycoprotein made up of two identical units, each consisting of three polypeptides: Aα (610 aa, 67 kDa), Bβ (461 kDa, 56 kDa) and γ (411 aa, 48 kDa). The main role of fibrinogen is its involvement in the blood coagulation cascade, where plasma fibrinogen is converted into an insoluble fibrin clot in the presence of thrombin. The cleavage sites for thrombin are located in the E-region, the central part of molecule formed by all six chains [1]. Within the fibrinogen sequence, lysine residues are located in close proximity to thrombin cleavage sites and polymerization motifs; fibrin cross-linking is accomplished by the formation of covalent bonds between glutamine and lysine residues within the α- and γ-chains in the presence of factor XIII (FXIII) [2]. Generally, fibrin formation and structure may be affected by a plethora of factors, including both nucleotide polymorphism(s) and posttranslational modifications, including limited proteolysis, alterations of *N*-glycosaminoglycans, amino acid phosphorylation, tyrosine sulfation, glycation, nitration and acetylation [3].

Under hyperglycemic conditions, which are inseparable from pathophysiological background occurring in patients with diabetes mellitus (DM), the extracellular and intracellular proteins are modified [4,5]. Glycation is the most common major non-enzymatic mechanism in DM, which proceeds through many stages. Primarily, glycation occurs at lysine residues with formation of glucosylamines (aldimines; Schiff bases) and fructosamines (ketoamines; Amadori compounds), which are named early glycation products. Subsequent irreversible reactions, which include crosslinks, formation of aromatic heterocycles and oxidized compounds, lead to production of advanced glycation end products (AGEs). Protein glycation is possible in the presence of glucose, and occurs independently of glucose concentration; however, the glucose level strongly determines the rate of glycation. Additionally, increased glucose concentration induces oxidative stress and the generation of highly reactive products, such as methylglyoxal (MGO), which is known to induce structural modifications and functional impairments of various proteins, including fibrinogen [6]. It has been observed that the glycation level in the fibrinogen molecule was 2–3-fold higher in type 2 diabetes mellitus (T2DM) than in non-diabetic individuals [4,5]. The changes in the level of fibrinogen glycation in DM patients were correlated to glycemic control monitored as a level of HbA_1c_ [7]. Indeed, fibrinogen has been shown to be glycated in vivo and in vitro and this modification may influence its interaction with other coagulation/fibrinolysis proteins. The plasma of the subjects with DM form fibrin clots with denser structure and enhanced resistance to fibrinolysis than those of healthy control subjects. In the cross-linking of the fibrin network by FXIII, as well in the binding to fibrin of various proteins involved in the lysis of the fibrin, the lysine residues are involved. It is quite plausible that binding of glucose at these sites may have a significant effect on the functionality of fibrinogen, specifically revealed as an altered fibrin network structure [5,7].

Although much attention has been devoted to the impairment of fibrinolysis in diabetes, fewer studies have addressed the role of fibrinogen modification in the process of fibrin polymerization [8,9]. An understanding of the mechanisms of fibrin polymerization is important for clinical medicine for a number of reasons; for example, many clinical assays in hospital coagulation laboratories measure gelling time to diagnose a whole array of disorders. The aim of the present study was to further clarify the nature of fibrin polymerization in hyperglycemic conditions by comparing a few variables describing its kinetic characteristics in plasma samples with standardized fibrinogen protein (antigen) concentration taken from T2DM and control subjects. We hypothesized that the ability of glycated fibrinogen to form a stable clot differs between the T2DM and non-diabetic subjects, and that this property influences the outcome of some diagnostic assays. Our findings demonstrate for the first time that the impaired clotting ability of glycated fibrinogen can influence the results of the standard Clauss diagnostic method, routinely used to determine fibrinogen concentration in plasma. The study also compares the immediate and delayed effects of glucose on the polymerization of fibrin(ogen).

## 2. Materials and Methods

### 2.1. Materials

Fibrinogen from human plasma (contains ≥90% clottable proteins) was obtained from Calbiochem (Darmstadt, Germany). Fibrinogen Human SimpleStep ELISA™ Kit was from Abcam (Cambridge, UK). Human thrombin (Chrono-Par) was from Chrono-Log (Havertown, PA, USA). Deoxymorpholinolinofructose (DMF), Zwittergent 3–14, nitroblue tetrazolium (NBT), BCA kit, glucose were and other reagents (HEPES, TRIS) from Sigma Aldrich (St. Louis, MO, USA). Glycated Serum Protein Assay (GlycoCap) was from Diazyme Laboratories (Poway, CA, USA). Mouse anti-CML monoclonal antibody was purchased from Abcam (Cambridge, MA, USA) and secondary anti-IgG horseradish peroxidase-conjugated polyclonal antibody was from Santa Cruz Biotechnology (Dallas, TX, USA).

### 2.2. Study Population and Blood Collection

The study enrolled 27 patients with T2DM, diagnosed based on the WHO criteria, and 22 donors without DM as a control (Table 1). The exclusion criteria for both groups were as follows: infection, recognized cancer disease, anemia, coronary artery disease, chronic inflammatory disease, liver disease, pregnancy, arterial or venous thromboembolic events within the preceding six months, or current anticoagulant therapy, all the states known to substantially alter blood coagulation or fibrin clot structure. The study was performed under the guidelines of the Helsinki Declaration for human research and approved by the Medical University of Lodz Committee on the Ethics of Research in Human Experimentation (approval number RNN/122/15/KB). All enrolled subjects and patients provided written informed consent. All participants underwent a comprehensive physical examination.

For every participant, the clotting parameters (thrombin time, prothrombin time, activated partial thromboplastin time (APTT)), lipidogram (total cholesterol, triglycerides, high-density lipoproteins (HDL) cholesterol), aspartate and alanine aminotransferase (AST, ALT) activities and fasting glucose were measured using standard diagnostic tests. The fibrinogen concentration in the samples of citrated plasma was determined with the use of two approaches: either with the standard Clauss method (as the diagnostic test) or by using the immunoenzymatic assay (Fibrinogen Human SimpleStep ELISA™ Kit; measurements performed according to the manufacturers’ procedure). The level of the glycated hemoglobin (HbA_1c_) was evaluated with immunoturbidimetric assay (ITA, Beckman Coulter, Brea, CA, USA).

The fibrinogen antigen concentration and other measurements (fibrin polymerization, fructosamine concentration, and fibrinogen isolation) were taken in platelet-poor plasma prepared from a whole blood collected into the tubes containing 3.2% sodium citrate (9:1). After withdrawal, whole blood was centrifuged (3000× *g*, 15 min), and plasma samples were collected and stored at −80 °C until used.

### 2.3. Fibrinogen Incubation with 30 mM Glucose under In Vitro Conditions

Purified human fibrinogen (Calbiochem; Darmstadt, Germany) (5 mg/mL in phosphate-buffered saline (PBS), pH 7.4, sterile conditions) was incubated with glucose (a final concentration of 30 mM) for four days (37 °C, mixing 300 rpm). After incubation, the samples of fibrinogen were frozen until further measurements or were dialyzed against HBS (20 mM HEPES, 0.154 M NaCl, pH 7.4) to remove any unbound glucose. Dialysis was performed in three steps: two steps against fresh sample buffer for 1.5 h dialysis at RT, and overnight dialysis at 4 °C. Protein concentration in the fibrinogen samples was measured by using BCA kit from Sigma Aldrich (St. Louis, MO, USA).

### 2.4. Turbidity Measurement of Fibrin(ogen) Polymerization

Before the measurements of fibrin polymerization, the concentration of fibrinogen antigen in the citrated plasma samples was determined using Fibrinogen Human SimpleStep ELISA™ Kit (Abcam, Cambridge, UK). Aliquots of citrated plasma were diluted in HBS buffer (20 mM HEPES, 0.154 M NaCl, pH 7.4) to reach the fibrinogen antigen concentration of 1 mg/mL in the reaction mixture, preincubated at 37 °C for 5 min and placed (240 µL) into the microplate wells to initiate the reaction in the presence of CaCl_2_ (5 mM) by the addition of 60 µL of human thrombin at a final concentration of 0.3 U/mL. The turbidity was monitored every 10 s for 15 min. The curves were characterized by two parameters: the maximal velocity (V_max_), calculated as the slope of the steepest part of the polymerization curve (using 4 time points), and the absorbance change (∆Abs, F_max_) over 15 min, calculated as a difference between the maximum and the baseline values of the absorbance. The maximal velocity (V_max_) represents the rate of lateral protofibril association, and the absorbance change (∆Abs, F_max_) indicates fibrin density in a clot. The reaction was monitored at 405 nm in a 96-well microplate reader (Benchmark, Bio-Rad, Hercules, CA, USA) at 37 °C.

The fibrin polymerization was also analyzed in the samples of purified human fibrinogen incubated with 30 mM glucose for four days. After incubation, or following dialysis (see above), the samples of fibrinogen were diluted in HBS buffer at a final concentration of 1 mg/mL, and measurements were taken for 15 min. Optionally, the determination of fibrin polymerization was evaluated in the samples of purified fibrinogen in the presence of 6 mM glucose: the glucose was added to the fibrinogen solution immediately before the measurements. Likewise, the maximal velocity (V_max_) and the absorbance change (∆Abs, F_max_) over 15 min reaction were calculated for every polymerization option.

### 2.5. Determination of Fructosamine Concentration in Plasma

The concentration of fructosamine, the product of a non-enzymatic reaction between glucose and amino acid residues of proteins, was measured in the citrated plasma using Glycated Serum Protein Assay (Diazyme Laboratories, Poway, CA, USA), according to the manufacturer’s protocol. The amount of fructosamine (µM) in plasma samples was expressed as units per mg plasma protein (protein determined by bicinchonic acid protein assay kit, BCA, Sigma, Saint Louis, MO, USA).

### 2.6. Fibrinogen Purification

Fibrinogen was isolated from the citrated plasma samples from T2DM patients and control subjects according to Dietrich et al. with some modifications [10]. The deep-frozen plasma samples were unfrozen and cooled to 4 °C. Following this, one part of 99% ethanol (100 µL) was added to nine parts of plasma (900 µL) and the mixture was intensively vortexed. After incubation (−20 °C, 20 min) the samples were centrifuged (4000× *g*, 20 min, 4 °C), and the pellets were washed two times with deionized water and re-centrifuged (4000× *g*, 3 min, 4 °C). In the end, the washed pellets were solubilized in TBS buffer (50 mM TRIS, 150 mM NaCl, pH 7.4). Fibrinogen yields were quantified spectrophotometrically (coefficient of extinction 1.55, absorption wavelength 280 nm). Purified fibrinogen was run on 8% SDS-PAGE gels to confirm purity and the absence of fibrinogen degradation products (Appendix A). No bands, apart from the three intact fibrinogen chains, were present on the gels (68 kDa for the α chain, 54 kDa for the β chain, and 48 kDa for the γ chain). The results were compared to those of commercial human fibrinogen purchased from Calbiochem (Darmstadt, Germany) with the use of a standard protein marker (Thermo Fisher Scientific PageRuler Prestained Protein Ladder, Waltham, MA, USA).

### 2.7. Analysis of Fibrinogen Glycation with Colorimetric NBT Assay

The level of glycated fibrinogen in control and diabetic plasma samples was assayed according to Gugliucci et al. [11] with some modifications. Briefly, 100 µL of standards or samples containing 1 mg/mL of fibrinogen was added to 150 µL of nitroblue tetrazolium (NBT) reagent (one part of 1 mM NBT in 0.2 mM carbonate buffer pH 10.35 mixed with one part of 10 mg/mL Zwittergent 3–14 in 0.2 mM carbonate buffer pH 10.35), and the mixture was incubated at 56 °C for 20 min. After incubation, the absorbance at 530 nm against a reagent blank was measured. Standard (deoxymorpholinolinofructose, DMF) was freshly prepared in the carbonate buffer. The degree of glycation was normalized for the protein concentration and the amount of glycated fibrinogen was then expressed as nmol of DMF equivalents per mg of protein.

### 2.8. Measurements of Carboxymethyllysine (CML) Level in Fibrinogen Using Western Blot

*N*-(6)-Carboxymethyllysine (CML), an advanced glycation end product formed in proteins by a combined nonenzymatic glycation and oxidation (glycoxidation) reactions, was detected in the fibrinogen samples isolated from T2DM and control plasma. Fibrinogen was isolated as described above. In the first step, the isolated fibrinogen (2 µg of protein per well) was separated into the three subunits by SDS-PAGE. After electrophoresis, the protein was transferred to a nitrocellulose membrane (200 mA, 2 h, 4 °C) and stained with Ponceau red (loading control). Next, the membrane was blocked with 5% BSA in TBS (25 mM TRIS, 154 mM NaCl, pH 7.4) for one hour at RT and washed with TBST (25 mM TRIS, 154 mM NaCl, 0.05% Tween 20, pH 7.4). After washing, the membrane was incubated overnight with monoclonal mouse anti-CML antibodies (1:400) at 4 °C, and then with polyclonal horseradish peroxidase-conjugated anti-IgG (1:10,000) for one hour at RT. The presence of CML in fibrinogen samples was detected with ECL substrates (Thermo Fisher Scientific Inc., Waltham, MA, USA). The images were acquired and analyzed using a ChemiDoc MP System (Bio-Rad, Santa Rosa, CA, USA).

### 2.9. Determination of Glycation Sites in Fibrinogen Molecule by LC MS/MS

Gel electrophoresis was applied before MS analysis to purify fibrinogen isolated from plasma samples (see Section 2.6 Fibrinogen isolation in M and M) and to identify the fibrinogen chains. Selected protein bands excised from gel slices were crushed into small pieces and subjected to in-gel tryptic digestion. Briefly, the gel pieces were first de-stained in acetonitrile/50 mM NH_4_HCO_3_, pH 8.0, (1:1, *v*/*v*) and then dehydrated in acetonitrile. Gel fragments were subjected to the reduction with 10 mM dithiothreitol for 30 min at 56 °C and alkylation with 50 mM iodoacetamide for 45 min at 25 °C in the dark, followed by two alternating washing steps, each with 25 mM NH_4_HCO_3_ and acetonitrile. Gel pieces were then dehydrated with acetonitrile, dried and subsequently rehydrated with a minimum volume of 25 mM NH_4_HCO_3_, pH 8.0, containing enough trypsin (sequencing grade, Promega) to provide a 1:10 trypsin-to-protein ratio. The incubation was performed overnight at 37 °C. After proteolysis, the supernatant was collected in PCR-tubes, while gel pieces were subjected to two further extraction steps with 70 mL of 0.1% trifluoroacetic acid (TFA) in 2% acetonitrile. After centrifugation, the supernatant was collected and subjected to ESI-MS/MS analysis (the conditions of MS analysis were described before [12]). The nanoflow ESI source conditions were set as follows: the capillary voltage was set to 1850 V, the sample cone voltage 40 V, the extraction cone voltage 5 V. Data acquisition was controlled by MassLynx™ 4.1 software (Waters). The sequence coverage was 70% for the α-chain, 46% for the β- and 72% for the γ chain. The obtained MS/MS spectra were searched against SwissProt database with Protein Lynx Global Server Software (PLGS version 2.5.3 Waters Corporation, Milford, MA, USA). The search criteria included glycation on lysine residues recognized as the correct parent ion mass shift values (fructosyl-lysine-2H_2_O (FL-2H_2_O) (+126.032 Da).

### 2.10. Statistical Analysis

Data were expressed as mean ± SD or median and interquartile range (IQR), i.e., lower (25%) to upper quartile (75%), depending on data distribution (Shapiro–Wilk’s test) and variance homogeneity (Brown–Forsythe’s test). The Student’s *t* test for independent samples or the Mann–Whitney U test were used to compare the groups (T2DM vs. control), depending on whether the data met the assumptions of data normality and variance homogeneity (either raw or Box–Cox-transformed data were used for the analyses). Two-way ANOVA with post hoc Tukey’s HSD test was used to analyze the differences in fibrinogen concentration between T2DM and control groups measured with two methods (Clauss or antigen in ELISA). The GLM ANCOVA was used as the inference test to adjust the compared variable for confounders (age, sex). The χ2 test was used to compare categorical variables. To analyze the statistical significance of the differences between fibrinogen samples incubated with or without glucose, the Wilcoxon singed rank test was used. Simple correlations were estimated with either rank Spearman correlation test or Kendall’s tau correlation (analyses for both T2DM and control groups), and the partial Spearman’s rank test adjusted for HbA_1c_ values (analyses for all participating groups and separately for T2DM patients). In testing statistical significance, two-tailed tests were used and a *p* value less than 0.05 was considered significant. All inference and association analyses used bootstrap-boosted approaches (10,000 iterations) to minimize the risk that any possible differences or associations could be claimed by a pure chance. Likewise, the bootstrap-boosted permutations were employed to estimate the probability of glycation at given Lys(K) residues in three fibrinogen chains. The lysine residues with the highest vulnerability to non-enzymatic glycosylation in each fibrinogen chain were predicted by NetGlycate software (http://www.cbs.dtu.dk/services/NetGlycate/) and NetGlycate server v1.0 (Technical University of Denmark) [13]. The agreement between the theoretical most preferable positions of glycation (estimated with NetGlycate) and the revealed positions of glycation in fibrinogen samples from diabetic patients and control subjects was analyzed with ROC analysis. Mountain plots, which show the distributions of the differences between two groups and are complementary plots to the difference plots [14], were used to demonstrate similarities in glycation sites between diabetic and control subjects. The likelihoods of glycation (more strictly, their ±95% ranges) were evaluated based on the NetGlycate scores normalized to a scale from 0 to 1, and the bootstrap-boosted frequencies of their occurrence in the examined groups of subjects, calculated with the use of the resampling adjusted for the sample size of the overall examined population [*n* = 22 + 27 = 49] [15]. Statistical analyses were performed using Statistica software v13.1 (Statsoft, Cracow, Poland) and Resampling Stats™ for Excel v4.0.

## 3. Results

### 3.1. Clinical Characteristics of Study Participants

The clinical characteristic of T2DM patients and control donors is shown in Table 1. In nine patients, the duration of diabetes exceeded five years, while diabetes had been recognized in the last year in the other nine. The remaining group of patients (9) suffered from diabetes longer than one year and shorter than five years. The T2DM group demonstrated significantly higher fasting glucose, HbA_1c_ fraction and fructosamine concentration, as well as higher BMI values. Additionally, the triglyceride concentration was significantly elevated for T2DM group, but total cholesterol and LDL cholesterol concentrations were not significantly different between groups. Diabetic peripheral neuropathy was observed in eight T2DM patients (30%), diabetic kidney disease in four T2DM patients (15%), and diabetic retinopathy in five patients (18%). The T2DM patients were treated with various anti hyperglycemic drugs (15 patients used insulin with various dosing regimens, either with or without other antihyperglycemic drugs, 7 patients were treated with metformin monotherapy, 4 with sulfonylourea monotherapy and one person was treated with the combination of metformin and gliclazide).

### 3.2. Clot Formation in Plasma Samples from T2DM Patients and Control Subjects

The representative curves for the polymerization of fibrinogen in the plasma samples are presented in Figure 1A. The analysis of clot formation in plasma diluted down to the fibrinogen concentration of 1 mg/mL revealed significantly lower maximal velocity (V_max_) in the T2DM samples (*p* < 0.05, Figure 2A). No significant difference in maximal absorbance change (F_max_) was found between the T2DM and control group (Figure 2B). In the course of fibrinogen polymerization ongoing in plasma, glucose was present in the range from 0.81 to 7.45 mM, nevertheless, glucose concentration was not revealed to be significantly associated with the maximal velocity (V_max_).

In the T2DM group, activated partial thromboplastin time (APTT) was significantly shorter (*p* < 0.001), but statistically equal thrombin time (TT) and prothrombin time (PT) values were observed in T2DM and control subjects (Table 2). No significant association was observed between APTT, TT or PT and fasting glycaemia, plasma fructosamine or HbA_1c_ in the T2DM group or the overall group. Plasma fibrinogen concentration, determined either by Clauss method or ELISA measurements of fibrinogen antigen, was higher for T2DM patients than control individuals (Figure 3; *p* < 0.05 for fibrinogen antigen). Additionally, the results obtained by the Clauss method were statistically higher than those of the fibrinogen antigen in both groups of participants (*p* < 0.001).

In the T2DM group and in the overall group comprising all study participants (T2DM + control), the maximal velocity (V_max_) was significantly positively associated with maximal absorbance (F_max_), and negatively with plasma fructosamine (Table 3). Additionally, fasting glycaemia may influence the velocity of fibrin polymerization (V_max_), but this relationship was not statistically significant (Table 3).

### 3.3. Clot Formation in Fibrinogen Samples Incubated In Vitro with Glucose

The incubation of human fibrinogen with glucose and the presence of glucose in the reaction mixture during polymerisation influenced its clotting ability (Figure 1B). A significantly lower maximal velocity (V_max_) was observed for the fibrinogen samples incubated for four days with glucose (V_max_ for the samples with glucose was 0.052 ± 0.033 AU/s vs. 0.060 ± 0.033 AU/s for control without glucose, *p* < 0.05, *n* = 6). The same tendency was observed in the fibrinogen samples supplemented with 6 mM glucose just before measurements (V_max_ for the samples with glucose was 0.052 ± 0.013 AU/s vs. 0.061 ± 0.009 AU/s for control without glucose, *p* < 0.05, *n* = 4). The samples incubated with glucose presented lower averaged differences in maximal absorbance change (F_max_) values (F_max_ = 0.308 ± 0.222 AU for glucose vs. F_max_ = 0.353 ± 0.250 AU for control, *p* = 0.17). However, rather huge variability resulted in no statistically significant differences in maximal absorbance change (F_max_) revealed between groups.

The changed clot structure, the increased clot weight and its decreased ability for retraction was observed in the presence of 30 mM glucose (Appendix A), but these differences were beyond a statistical significance.

Otherwise, the polymerization curves did not significantly differ for the samples of fibrinogen incubated with 30 mM glucose and dialyzed (glucose was absent in the reaction mixture during polymerization). The V_max_ for the samples incubated with glucose was 0.094 ± 0.023 vs. 0.105 ± 0.010 AU/s for control samples (fibrinogen incubated without glucose), (*p* = 0.116, *n* = 6). Colorimetric NBT assay showed significantly higher levels of fuctosamine in the samples incubated with 30 mM glucose and then dialyzed (48.7 ± 12.9 nmol DMF/mg protein) than control samples (35.6 ± 4.7 nmol DMF/mg protein, *p* < 0.05, *n* = 6), indicating greater fibrinogen glycation. The significant negative association was observed between fibrinogen glycation level and V_max_ (R***_S_*** = −0.872, *p* < 0.05), while positive between fibrinogen glycation level and F_max_ (R***_S_*** = 0.810, *p* < 0.05).

### 3.4. Extent of Fibrinogen Glycation Evaluated with Various Analytical Assays

In the samples of fibrinogen isolated from diabetic and control plasma, the level of non-enzymatic glycosylation (glycation) was analyzed using two methods: (1) determination of fructosamine by the reduction of nitroblue tetrazolium (NBT), (2) detection and quantification of carboxymethyllysine (CML) by Western blotting (Appendix A). It was observed that fructosamine level in the NBT assay was higher in diabetic fibrinogen (28.6 ± 15.5 nmol DMF/mg protein) compared to the control (22.6 ± 11.2 nmol DMF/mg protein; *p* = 0.253), but this difference was not statistically significant. Additionally, the amount of CML in the α, β, γ fibrinogen chains, measured separately and together, did not differ between the groups: the total CML level in fibrinogen samples expressed as a mean chemiluminescence detected with anti-CML antibody per protein amount was 0.57 ± 0.16 for T2DM vs. 0.60 ± 0.16 for control, *n* = 8 for each group.

### 3.5. Structural Study of Fibrinogen Glycation

The presence of fructosyl-lysine (FL), an early product of glycation resulting from the spontaneous reaction between glucose and lysine in protein, was identified in fibrinogen isolated from T2DM patients and control subjects by peptide sequencing using LC MS/MS. The β and γ chains of fibrinogen contain greater proportions of lysine residues (34/453 and 36/491 amino acids, resp.) than the α chain (43/866 amino acids). However, NetGlycate server evaluation found that merely 1.73% of Lys residues in the α chain, 3.05% in the β chain and 1.77% in the γ chain were vulnerable to glycation (Figure 4, Figure 5 and Figure 6). Noteworthy, in all cases the γ chain in T2DM, the potentially glycated positions, as evaluated by NetGlycate, did not match significantly the really glycated positions of Lys residues found in the α, β and γ chains of human fibrinogen, either in control volunteers or T2DM patients: ROC_AUC(+SE) diabetic_ = 0.611 + 0.094 (n.s.) and ROC_AUC(+SE) control_ = 0.564 + 0.095 (n.s.) for the α Fg chain, ROC_AUC(+SE) diabetic_ = 0.595 + 0.099 (n.s.) and ROC_AUC(+SE) control_ = 0.438 + 0.097 (n.s.) for the β Fg chain and ROC_AUC(+SE) diabetic_ = 0.740 + 0.103 (*p* < 0.02) and ROC_AUC(+SE) control_ = 0.548 + 0.121 (n.s.) for the γ Fg chain.

In the case of all three fibrinogen chains in both T2DM patients and control subjects, glycation occurred at both the sites of theoretically higher glycation potential (evaluated according to the NetGlycate algorithm) and the sites of theoretically low glycation potential, i.e., where glycation was much less likely (Table 4). Diabetic patients demonstrated a considerable proportion of sites with lower glycation (ranging from 0.4 to 0.6 and particularly pronounced in gamma chains), and this number was even greater in control subjects (a proportion of up to 0.8 and was particularly pronounced in the beta and gamma chains). More interestingly, the glycation sites identified in the chains of fibrinogen samples originating from T2DM patients and control subjects did not correspond to each other, as demonstrated on the mountain plots (Figure 7, Figure 8 and Figure 9). As shown, all chains demonstrated significant differences between control subjects and diabetic patients with regard to the glycated positions: the curves in these are clearly skewed to the right, since the number of glycation sites in diabetic patients always exceeded those detected in control subjects. The asymmetry around zero, when comparing control and diabetic subjects with the NetGlycate designs, further confirms the above-mentioned discordance between theoretical and practical estimates.

## 4. Discussion

The present study was undertaken to determine whether fibrin(ogen) polymerization is influenced by hyperglycemia. Although previous studies have demonstrated changes in clot structure to occur in the case of diabetes mellitus, only limited data describing the consequences of fibrin(ogen) glycation on its clotting properties have been obtained [16]. Lutjens et al. did not observe any difference in fibrin polymerization between diabetic subjects and controls, but they showed that alpha chain crosslinking was impaired in the diabetic patients [9]: fibrinogen was found to be 35% more glycated in the diabetic patients, and a significant positive correlation was observed between the degree of glycation of fibrinogen and the defective alpha chain polymerization [9]. In an earlier study, Przygodzki et al. reported approximately 50% reduced initial plasma clotting velocity in diabetic rats, and that this decrease was present as soon as seven days after the onset of diabetes [17].

The present study was performed on fibrin clots formed in human plasma with the same level of fibrinogen antigen. It also analyzed fibrinogen polymerization after incubation with 30 mM glucose (hyperglycemic conditions) in two setups: (1) in the presence of glucose (non-dialyzed samples), and (2) in the absence of glucose (dialyzed samples). We hypothesized that the polymerization of fibrinogen may be influenced not only by its non-enzymatic glycosylation (glycation) per se, but also by the interactions of fibrinogen molecules with glucose molecules. Our findings confirm our hypothesis: significantly lower maximal velocity (V_max_) was observed in the T2DM plasma than the control plasma. Similar results were obtained from experiments with glucose-treated fibrinogen, where the V_max_ of the samples incubated with glucose was significantly lower than in samples without glucose. Furthermore, the addition of glucose (6 mM) to the reaction mixture with native fibrinogen reduced V_max_ of the relevant polymerization curves. Additionally, only a slight insignificant reduction of V_max_ was observed for the glucose-incubated fibrinogen when glucose was absent during polymerization, compared to control fibrinogen not subjected to the influence of glucose. Hood et al. have reported a reduced fiber overlap in a fibrin subjected to glucose, decreased fibrin length and fractal dimension, and increased porosity of fibrin incubated in media containing 10 mM glucose compared to glucose-free control media [18]. Native fibrinogen was incubated with glucose (6 or 10 mM) for 48 h, following which, a fibrin clot was polymerized in the presence of glucose. The extent of glucose incorporation into fibrinogen, determined as the difference between glucose concentrations recorded for time 0 and time 48 h, was found to be the highest under the hyperglycemic condition. Molecular docking and molecular dynamics simulation suggest that glucose adsorption on fibrinogen is highly dependent on the glucose concentration and characterized by a high affinity of glucose molecule to the hydrophobic amino residues [18]. In addition, Lippi et al. have note that fibrinogen concentration measured by the Clauss method significantly decreased with increasing glucose contamination (0–5–10–20%) [19]. The authors conclude that the specimens should not be analyzed if glucose contamination in samples designed for the analysis of hemostatic parameters is suspected or has been confirmed to occur [19]. It is noteworthy that glucose at the concentrations of 5%, 10% and 20%, corresponding—respectively, to glucose plasma levels of 19.2 mM (346 mg/dL), 33.2 mM (598 mg/dL) and 62.1 mM (1118 mg/dL), may be representative of a combined effect: the largest number of contaminated samples received in a conventional clinical laboratory, as well as of in vivo hyperglycemia frequently found in critical patients [19].

Furthermore, our findings may have implications on the future development of diagnostic approaches. To determine plasma fibrinogen concentration, several assays have been developed [20]. The most frequently used in a diagnostic routine is the Clauss assay: a method based on clotting time that measures the ability of fibrinogen to be enzymatically converted to a fibrin clot. In principle, diluted citrated plasma is activated with a high concentration of thrombin, and the clotting time is inversely proportional to the functional fibrinogen concentration. Stec et al. suggested that the functional intact fibrinogen test (FiF), a monoclonal antibody-based assay that measures the amount of fibrinogen antigen, may be a better overall predictor of the overall risk of cardiovascular disease [21]. The authors indicated that after adjustment to covariates, FiF remained significantly correlated with prevalent cardiovascular risk, whereas Clauss did not. Baker et al. found fibrinogen level to be higher in subjects with ischemic heart disease when assayed with the use of the nephelometric method, but not with the Clauss method [22]. A number of studies have reported higher plasma levels of fibrinogen in patients with type 1 diabetes and in type 2 diabetes [23]. In the present study, although ELISA analysis found fibrinogen antigen concentration to differ significantly between the T2DM and control groups, the Clauss assay only found fibrinogen concentration to be slightly elevated in T2DM. Although the present results are not conclusive, it is possible that the Clauss clotting time assay did not measure dysfunctional modified fibrinogen. As it was mentioned above, as well as previously suggested by others, only evaluation of fibrin polymerization using more accurate methods such as thromboelastometry or ELISA analysis were able to reveal fibrinogen deficiency and dysfunction. It is possible that the more intense stimulation by thrombin in the Clauss assays than in the tissue factor–initiated thromboelastometry, could mask the impact of dysfunctional fibrinogen [20]. Suzuki et al. have suggested that Clauss assays with a combination of clot waveform analysis allows the quantitative detection of fibrinogen disorder easily and represents a novel screening test for fibrinogen disorders [24].

Our present findings indicate significantly shorter activated partial thromboplastin time (APTT) in the T2DM group compared to control subjects, but without any significant differences in thrombin (TT) or prothrombin (PT) time. Lippi et al. reported that diabetic patients and those with impaired fasting glucose (IFG) (fasting glucose > 7 mM) had significantly shortened APTT [25]. In another study, Zhao et al. observed that whether patients with or without diabetes were grouped according to HbA_1c_ or fasting plasma glucose (FPG) levels, the APTT values in the diabetic, high-risk diabetic or impaired fasting glucose (IFG) groups were significantly shorter than in the euglycemic group, and APTT values below the reference range (APTT < 22 s) were more frequent [26]. Additionally, fibrinogen levels were significantly higher in the diabetic and high risk diabetic groups than in the euglycemic group, and fibrinogen values above the reference range (fibrinogen > 4.0 g/L) were more frequent in the diabetic groups. In our study HbA_1c_ level was a minimum of 7% for T2DM patient group, and was in the range of 4.9–6.1% for the control group, but the fasting plasma glucose concentrations widely varied (min-max 5.7–13.7 mM for T2DM, and 3.7–6.1 mM for control). Further investigations are needed to confirm whether a shortened APTT should be considered as a consequence of non-enzymatic glycosylation of coagulation factors, or whether it may be a result of the interactions between glucose itself and coagulation factors. Additionally, in our study no significant associations between coagulation parameters, such as APTT, PT, TT or kinetic characteristics of fibrinogen polymerization, and glycaemia or fructosamine concentration were found. It may be due to small group sizes and certainly needs further investigation. In our present study the demographic characteristics of the groups showed disproportions in sex (men are more abundant in control compared to diabetic group) notwithstanding, the statistical analysis of the investigated parameters after the adjustment for age and sex did not significantly modify the statistical significance of the analyzed inter-group differences.

The presence of glycated fibrinogen was confirmed in both the plasma from diabetes patients [7,11,27], and in the samples incubated in vitro with glucose [18,28]. Generally, several methods have been used to determine fibrinogen glycation, such as the hydroxymethylfurfural assay [27], Gly-Pro^®^assay [7] or NBT assay [11]. Our present findings identified an elevated glycation level (by about 30%) in diabetic fibrinogen or in fibrinogen incubated with glucose in vitro using the modified NBT assay [11]. Labelling experiments with [^14^C]glucose identified the presence of 0.12–0.5 mole bound glycated residues per mole fibrinogen when incubated with concentrations in the physiological range (i.e., 2–8 mM). In contrast, molar ratios of about 2 and 10 were found for samples incubated with 20 and 100 mM glucose [28]. These results are roughly in agreement with those of Mirshahi et al. [16], Austin et al. [4], Lütjens et al. [27] and Pieters et al. [7], who reported the extents in vitro and in vivo glycation of fibrinogen. It has been suggested that glucose adducts at two or possibly three positions found for glucose, would theoretically allow for maximal binding 4–6 mole of glucose/mole fibrinogen. Peptide sequencing of the glycated proteins and the identification of glycation sites was quite a challenging task—mainly due to differences in coverage yield in the analysis of proteins and the difficulty in establishing of the assessment criteria for glycation.

Quantitative assays of protein glycation are usually supplemented with results from mass spectrometry (MS) analysis. Svensson et al. identified two glycated lysines (K-133 in the β-chain and K-75, alternatively 85, in the *γ*-chain) in the in vitro glycated fibrinogen, and these are within the “plasmin-sensitive” coiled–coil regions [28]. In our experimental conditions and upon analyzing of several samples from patients with and without diabetes, we have noted that the LCMS/MS technique does not allow accurate quantitative comparison of the degrees of glycation between groups, as far as the process of glycation is non-specific. Differences in the detected glycation sites between repeats of LC MS/MS analysis of the same protein in different patients suggest that the individual copies of the protein molecules are not glycated in the same sites. We propose that it is rather a stochastic process, in which the overall extent of the modification is a function of reaction time and a concentration of glucose/fructose. It should be noted, however, that despite the lack of absolute specificity, the overall glycation of the protein may affect its conformation and hence, its activity.

It is worthwhile to mention that besides plasma soluble fibrinogen produced by liver, some amount of this protein is stored in blood platelets. It is secreted upon a strong activation of platelets with such stimuli as thrombin and collagen. Therefore, it may be assumed that it also takes part in thrombus formation under physiological conditions. It is also not clear whether this platelet pool of protein undergoes glycation under hyperglycemia. Having in mind that the ratio of platelet to plasma fibrinogen in a whole blood was determined to be 1:30 [29], we have focused in our study on the plasma soluble fraction of the protein. The role of platelet pool of fibrinogen and its plausible modifications in thrombus formation require further studies.

## 5. Conclusions

The clotting ability of fibrinogen may be determined by both non- enzymatic glycosylation (glycation), where new strong covalent bonds are formed, and by the presence of glucose at higher concentrations during fibrin polymerization, where energetically weaker interactions occur. The decreased maximal velocity (V_max_) of fibrin polymerization in T2DM plasma or in fibrinogen incubated with glucose seems to be the result of the combination of both fibrinogen glycation and interactions between glucose and fibrinogen molecules. Additionally, measurements of fibrinogen antigen concentration seem to be a more reliable and discriminative assay than optical methods based on fibrinogen clotting activity, such as the Clauss method, in the samples in which the fibrinogen structure could have become modified. The reduced clotting ability of glycated fibrinogen can influence the results of the standard clotting-based methods used to determine fibrinogen concentration in plasma.

## Figures and Tables

**Figure 1 biomolecules-10-00877-f001:**
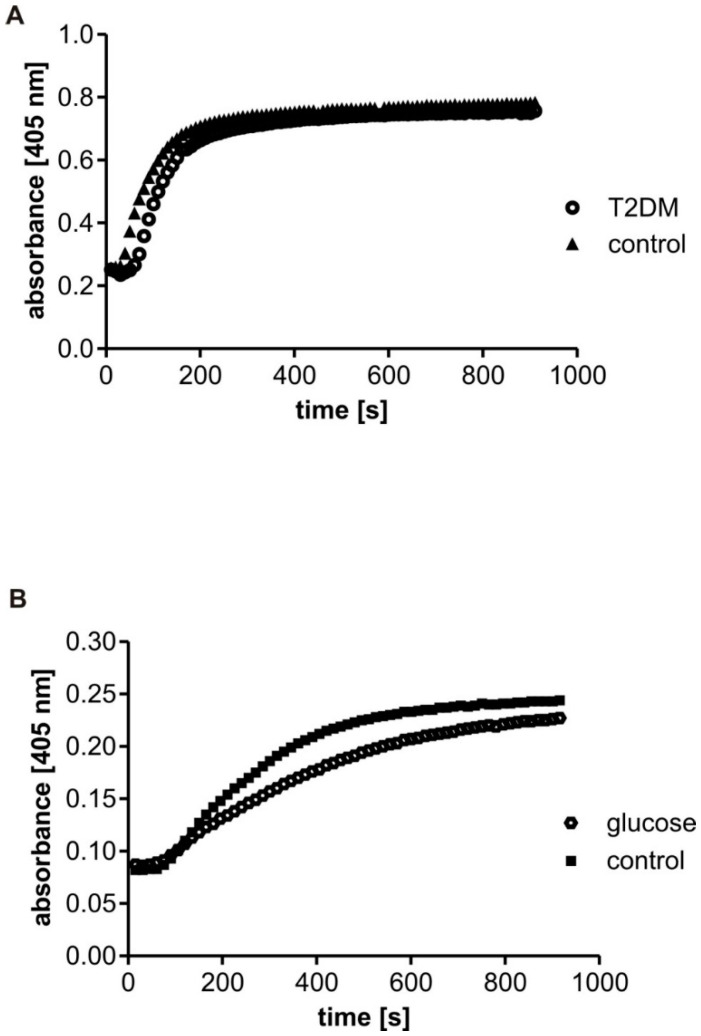
Representative curves of fibrin polymerization in plasma and in fibrinogen solutions Polymerization was monitored in plasma from control subjects (triangles) and diabetic patients (circles) (**A**) and in the solutions of fibrinogen incubated in vitro with PBS (squares) or 30 mM glucose for four days (37 °C, mixing 300 rpm/min)) (hexagons) (**B**).

**Figure 2 biomolecules-10-00877-f002:**
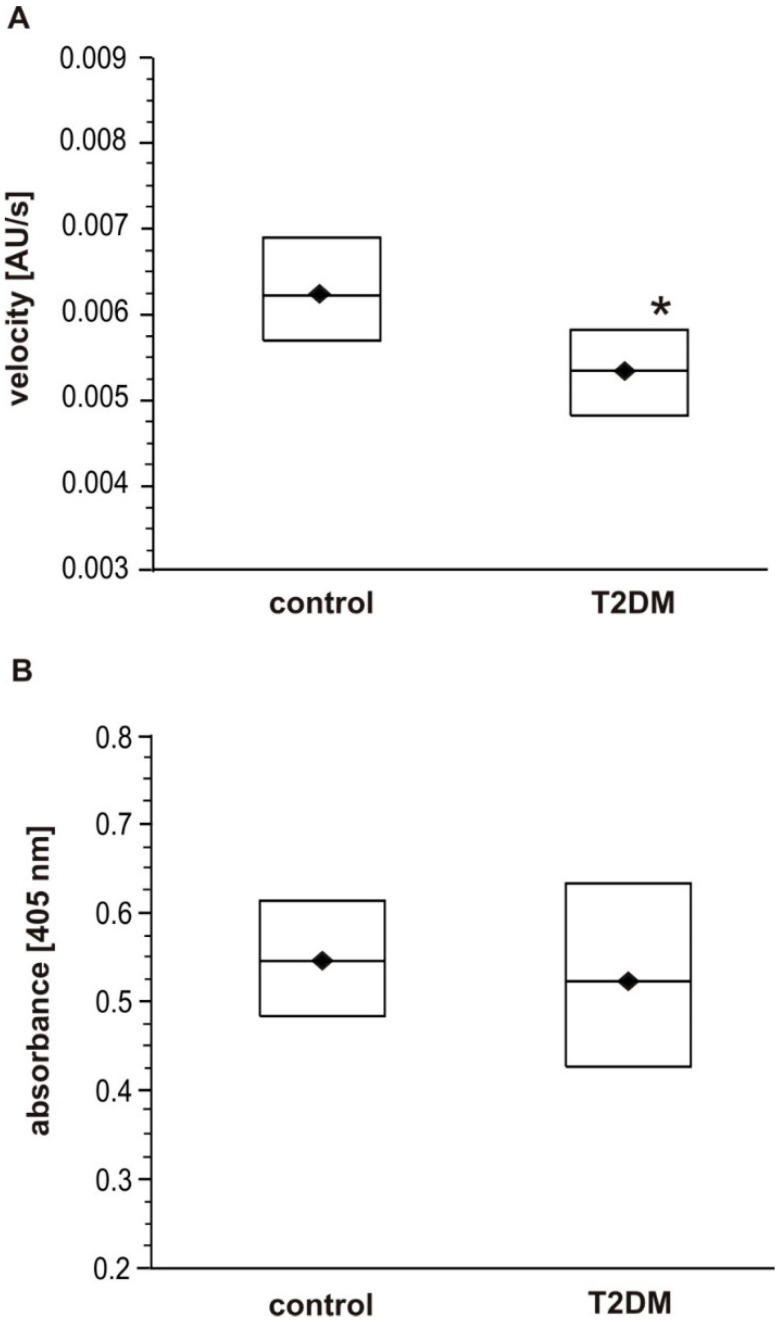
Fibrin polymerization in plasma from control subjects and T2DM patients: (**A**) maximal velocity (V_max_); (**B**) absorbance change (F_max_). Data shown as median (horizontal bar) and lower-upper quartile range (box); *n* = 24 for DM, *n* = 20 for control. The clot formation rate is represented by two variables: the maximal velocity (V_max_) (**A**) and the change in the absorbance (F_max_) calculated as a difference between absorbance maximum and baseline (**B**). Significance of differences estimated with the bootstrap-boosted Mann–Whitney U test. The V_max_ appeared significantly lower in T2DM patients (* *p* < 0.05). The absorbance change (F_max_) did not differ significantly between groups.

**Figure 3 biomolecules-10-00877-f003:**
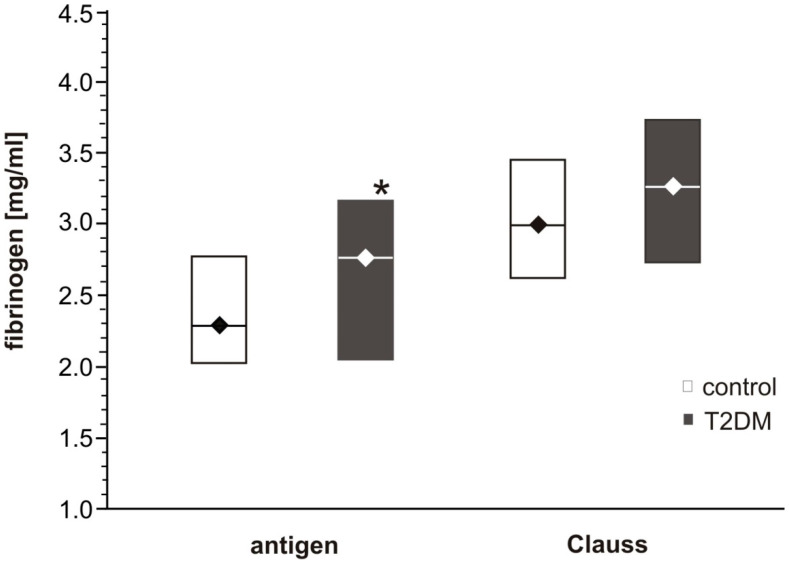
Fibrinogen concentrations in plasma from T2DM patients and control subjects. Data shown as the median (horizontal bar) and the lower–upper quartile range (box); *n* = 24 for DM, *n* = 20 for control. The concentration of fibrinogen antigen was significantly higher in DM plasma (grey boxes) compared to control plasma (white boxes) (* *p* < 0.05 by two-way ANOVA for repeated measures and the bootstrap-boosted Student *t* test).

**Figure 4 biomolecules-10-00877-f004:**
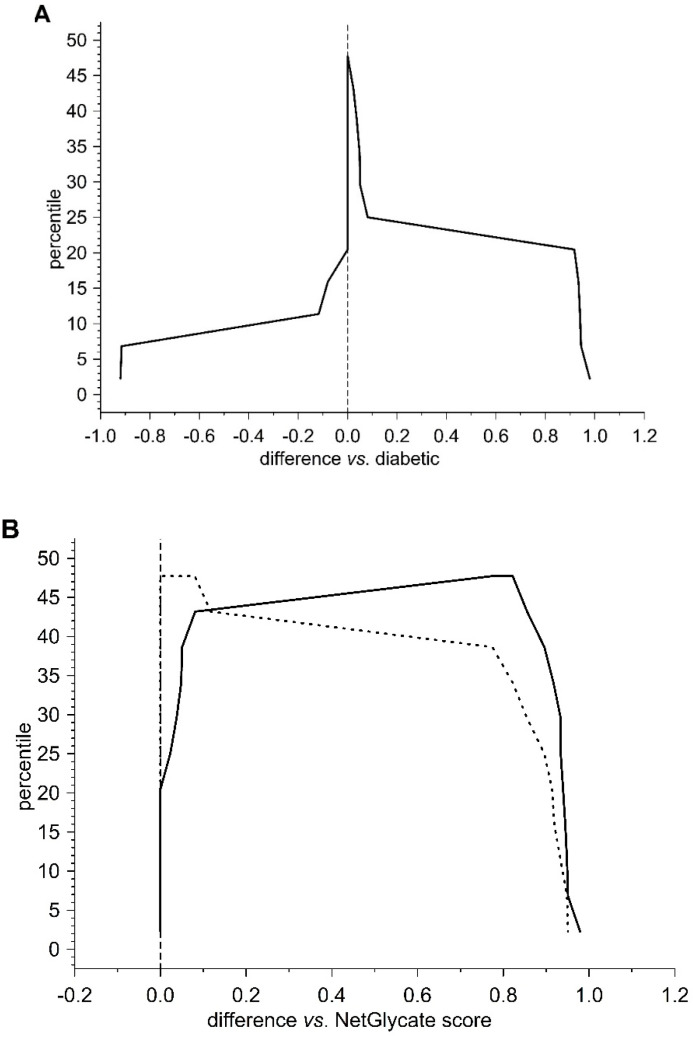
Mountain plots comparing the NetGlycate scores of glycation sites in control subjects and T2DM patients with theoretical estimates for glycation positions of high or low glycation potential in alpha chain of fibrinogen. The plot was created by evaluating the percentiles for ascending differences between the normalized scores: (**A**) differences of the NetGlycate-estimated scores (normalized to the scale from 0 to 1) between diabetic patients and control subjects (*norm score*_diabetic_ − *norm score*_control_), (**B**) differences of the NetGlycate-estimated scores (normalized to the scale from 0 to 1) between theoretical estimates for high or low glycation potential sites and the actual glycation sites revealed in diabetic patients (solid line) and control subjects (dashed line) (*norm score*_theoretical_ − *norm score*_control_; *norm score*_theoretical_ − *norm score*_diabetic_).

**Figure 5 biomolecules-10-00877-f005:**
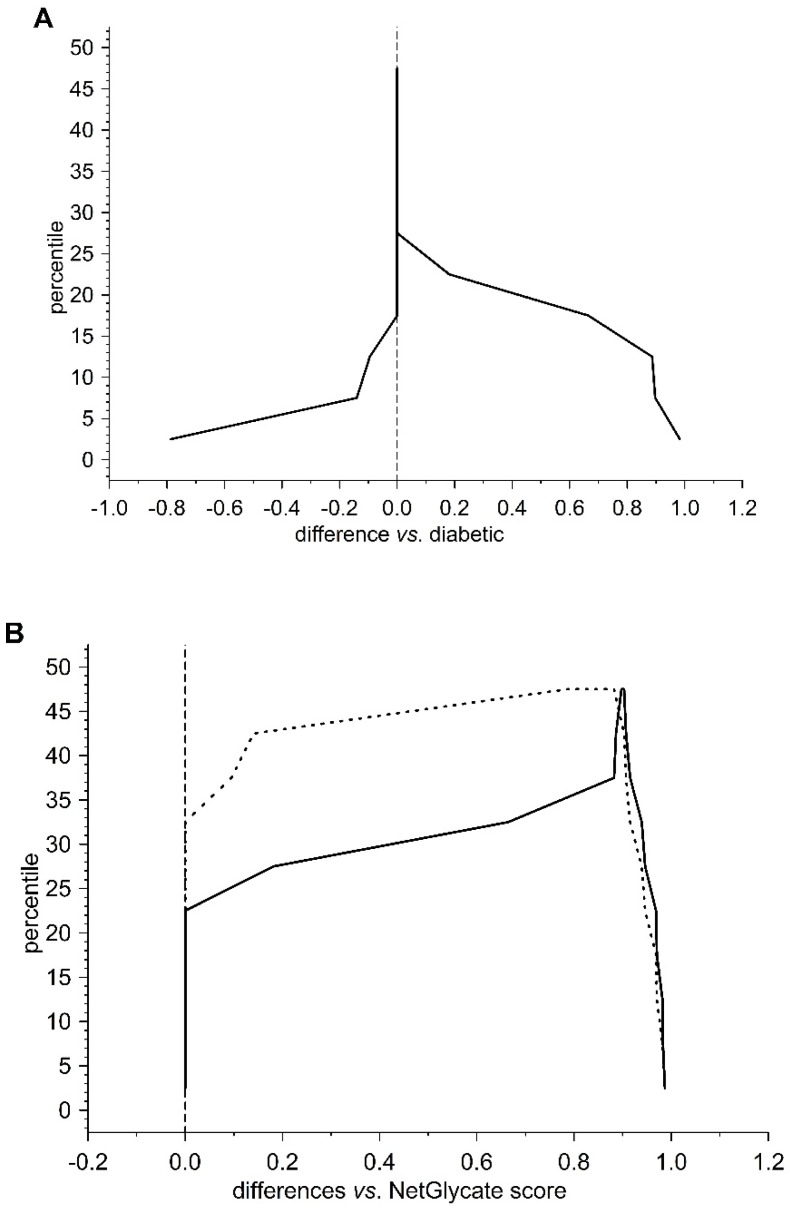
Mountain plots comparing the NetGlycate scores of glycation sites in control subjects and T2DM patients with theoretical estimates for glycation positions of high or low glycation potential in beta chain of fibrinogen. The plot was created by evaluating the percentiles for ascending differences between the normalized scores: (**A**) differences of the NetGlycate-estimated scores (normalized to the scale from 0 to 1) between diabetic patients and control subjects (*norm score*_diabetic_ − *norm score*_control_), (**B**) differences of the NetGlycate-estimated scores (normalized to the scale from 0 to 1) between theoretical estimates for high or low glycation potential sites and the actual glycation sites revealed in diabetic patients (solid line) and control subjects (dashed line) (*norm score*_theoretical_ − *norm score*_control_; *norm score*_theoretical_ − *norm score*_diabetic_).

**Figure 6 biomolecules-10-00877-f006:**
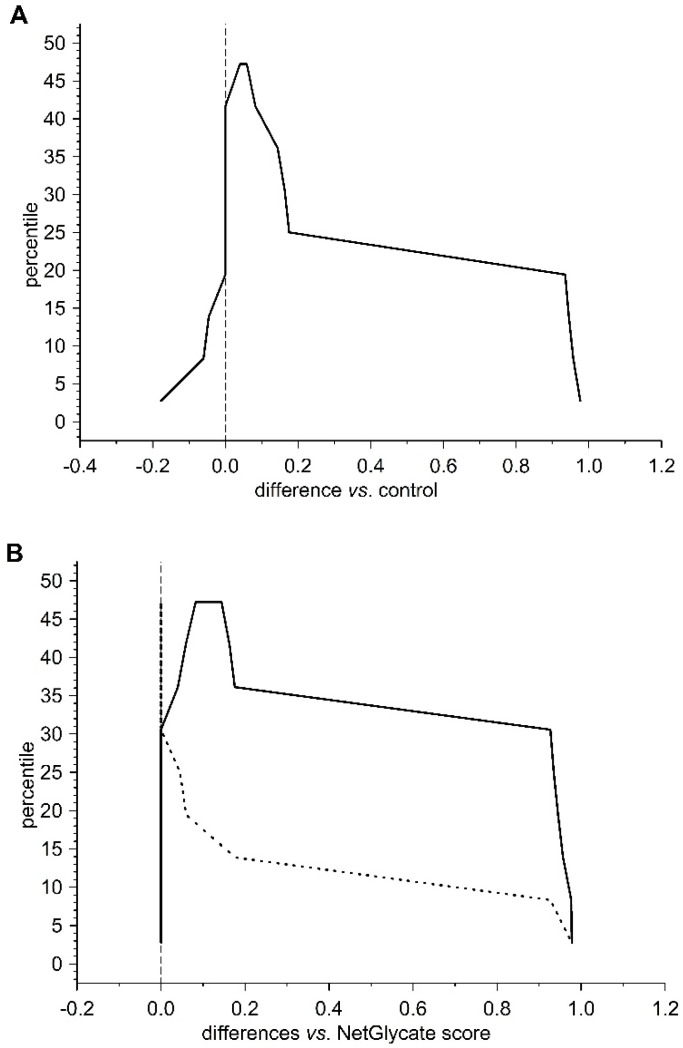
Mountain plots comparing the NetGlycate scores of glycation sites in control subjects and T2DM patients with theoretical estimates for glycation positions of high or low glycation potential in gamma chain of fibrinogen. The plot was created by evaluating the percentiles for ascending differences between the normalized scores: (**A**) differences of the NetGlycate-estimated scores (normalized to the scale from 0 to 1) between diabetic patients and control subjects (*norm score*_diabetic_ − *norm score*_control_), (**B**) differences of the NetGlycate-estimated scores (normalized to the scale from 0 to 1) between theoretical estimates for high or low glycation potential sites and the actual glycation sites revealed in diabetic patients (solid line) and control subjects (dashed line) (*norm score*_theoretical_ − *norm score*_control_; *norm score*_theoretical_ − *norm score*_diabetic_).

**Figure 7 biomolecules-10-00877-f007:**
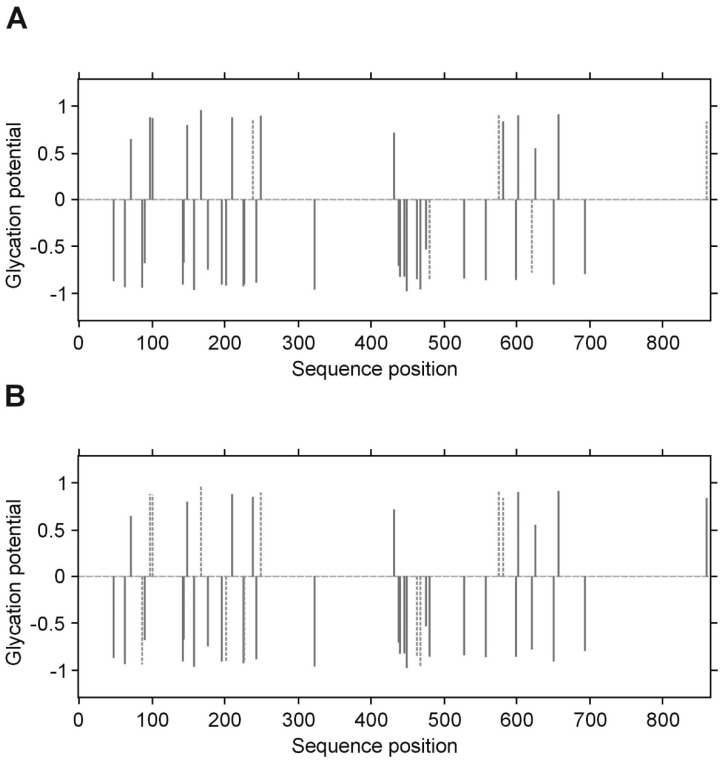
Predicted glycation sites in the α chain of human fibrinogen, evaluated by NetGlycate v1.0 software. Each plot presents the scores of glycation potential (bars), given for all 43 Lys residues of the α chain in control volunteers (**A**) and diabetic patients (**B**), which are relevant to the numbers between −1 and 1 (threshold marked as grey dashed line); a score above 0 means that the residue is a predicted glycation site, while a score below 0 means that the residue is unlikely to be glycated. The bars marked by dashed thick lines denote the residues that were found to be really glycated in the examined subjects.

**Figure 8 biomolecules-10-00877-f008:**
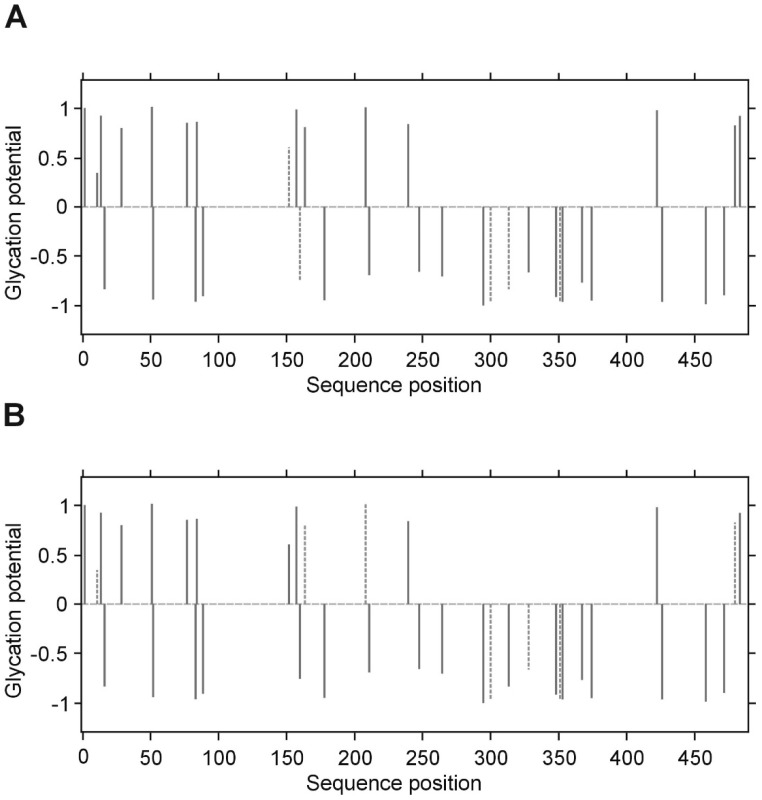
Predicted glycation sites in the β chain of human fibrinogen, evaluated by NetGlycate v1.0 software. Each plot presents the scores of glycation potential (bars), given for all 36 Lys residues of the β chain in control volunteers (**A**) and diabetic patients (**B**), which are relevant to the numbers between −1 and 1 (threshold marked as grey dashed line); a score above 0 means that the residue is a predicted glycation site, while a score below 0 means that the residue is unlikely to be glycated. The bars marked by dashed thick lines denote the residues that were found to be really glycated in the examined subjects.

**Figure 9 biomolecules-10-00877-f009:**
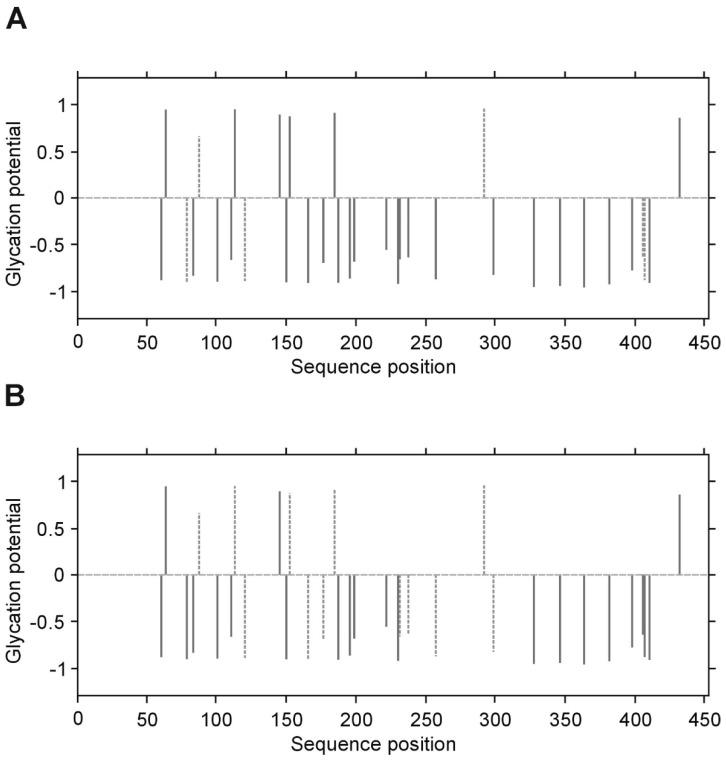
Predicted glycation sites in the γ chain of human fibrinogen, evaluated by NetGlycate v1.0 software. Each plot presents the scores of glycation potential (bars), given for all 34 Lys residues of the γ chain in control volunteers (**A**) and diabetic patients (**B**), which are relevant to the numbers between −1 and 1 (threshold marked as grey dashed line); a score above 0 means that the residue is a predicted glycation site, while a score below 0 means that the residue is unlikely to be glycated. The bars marked by dashed thick lines denote the residues that were found to be really glycated in the examined subjects.

**Table 1 biomolecules-10-00877-t001:** Characteristics of type 2 diabetes mellitus (T2DM) patients and control subjects.

Parameters	Control (*n* = 22)	T2DM (*n* = 27)	Statistical Significance
Age (years)	57.5 (45.8; 61.0)	60.1 (56.1; 63.0)	*p* = 0.084
Sex: men/women	14/8	9/16	ns
Fasting glucose (mmol/L)	5.16 ± 0.57	9.05 ± 2.42	*p* << 0.001
HbA_1c_ (%)	5.56 ± 0.34	9.56 ± 0.22	*p* << 0.001
Fructosamine (µm/mg protein)	315 ± 103	529 ± 240	*p* << 0.001
BMI (kg/m^2^)	24.8 ± 3.5	33.3 ± 7.0	*p* << 0.001
**Plasma lipids (mmol/L):**			
Total cholesterol	4.9 ± 1.3	5.2 ± 1.5	ns
Triglycerides	1.5 ± 0.8	2.2 ± 1.8	*p* < 0.02
HDL cholesterol	1.1 ± 0.5	1.2 ± 0.3	ns
LDL cholesterol	3.0 ± 1.1	3.0 ± 1.2	ns
**Liver markers (U/L):**			
ALT	21.5 (19.0; 36.9)	27.1 (20.0; 61.0)	ns
AST	26.0 (18.0; 42.0)	27.0 (14.0; 49.0)	ns
Systolic BP (mmHg)	120 (110; 140)	130 (130; 140)	ns
Diastolic BP (mmHg)	80 (70; 80)	80 (80; 80)	ns

Data shown as means ± SD or as medians and interquartile ranges (Q1:Q3). All the variables (either raw or after the Box–Cox transformations when the assumptions of normal distribution and/or homogeneity of variances were violated) were compared upon the adjustment for age and sex with the bootstrap-boosted (10,000 iterations) analysis of covariance (ANCOVA). Abbreviations: ALT: alanine aminotransferase/alanine transaminase; AST: aspartate aminotransferase/aspartate transaminase; BMI: body mass index; BP: blood pressure; HbA_1c_: glycated haemoglobin; HDL: high-density lipoproteins; LDL: low-density lipoproteins.

**Table 2 biomolecules-10-00877-t002:** The comparison of the selected coagulation parameters (thrombin time, prothrombin time, and activated partial thromboplastin time (APTT)) of T2DM patients and control subjects.

Parameters	Control (*n* = 22)	T2DM (*n* = 25)	Statistical Significance
Thrombin time (s)	18.3 ± 1.4	18.2 ± 1.5	ns
Prothrombin time (s)	13.5 ± 1.5	12.4 ± 1.0	ns
APTT (s)	33.0 ± 4.2	26.4 ± 4.0	*p* < 0.001

Data are shown as the mean ± SD. All the variables were compared upon the adjustment for age and sex with the bootstrap-boosted (10,000 iterations) analysis of covariance (ANCOVA). The statistical significance of differences between T2DM and control groups was analyzed using the bootstrap-boosted unpaired Student *t* test.

**Table 3 biomolecules-10-00877-t003:** The correlation coefficients (R***_S_***) for the maximal clot formation velocity (V_max_) and selected parameters in T2DM patients, and the overall group of examined subjects (control volunteers + T2DM patients).

Parameters	T2DM (*n* = 27)		Control + T2DM (*n* = 49)	
Glycaemia	−0.364	*p* = 0.062	−0.241	*p* = 0.09
fructosamine	−0.171	*p* = 0.395	−0.335	*p* < 0.05
F_max_	0.797	*p* < 0.001	0.775	*p* < 0.001

The association between the variables was estimated by the bootstrap-boosted simple Spearman rank test (T2DM group) or the bootstrap-boosted partial Spearman rank test adjusted for HbA_1c_ (overall group).

**Table 4 biomolecules-10-00877-t004:** The relevance of fructosyl-lysine residues in α, β and γ chains of fibrinogen from examined T2DM patients and control subjects to the positions of the NetGlycate-predicted glycation sites of high or low glycation potential.

Fg Chain		High Glycation Potential	Likelihood of Glycation	Low Glycation Potential	Likelihood of Glycation
alpha	control	238	0.222–0.279	480	0.019–0.024
		575	0.557–0.617	620	0.041–0.048
		859	0.571–0.627		
	T2DM	97	0.205–0.264	87	0.008–0.011
		100	0.433–0.502	202	0.011–0.014
		167	0.213–0.276	227	0.013–0.019
		249	0.206–0.266	463	0.018–0.023
		575	0.209–0.269	467	0.005–0.006
		581	0.200–0.258		
beta	control	152	0.191–0.239		
				160	0.034–0.042
				300	0.011–0.013
				313	0.023–0.029
				351	0.008–0.010
	T2DM	11	0.308–0.357	300	0.010–0.012
		163	0.194–0.249	328	0.040–0.051
		208	0.216–0.278	351	0.007–0.009
		479	0.195–0.252		
gamma	control	88	0.199–0.251	79	0.011–0.014
		292	0.235–0.296	121	0.011–0.016
				406	0.043–0.054
				407	0.015–0.018
	T2DM	88	0.182–0.234	121	0.010–0.013
		114	0.214–0.275	166	0.009–0.011
		153	0.205–0.263	177	0.032–0.041
		185	0.209–0.269	232	0.036–0.046
		292	0.214–0.275	238	0.038–0.049
				258	0.013–0.017
				299	0.018–0.023

The numbers in the columns ‘high/low glycation potential’ denote the positions of glycated Lys residues with high (likely to become glycated) and low (unlikely to become glycated) glycation potential. The likelihoods of glycation (more strictly, their ±95% ranges) were evaluated based on the NetGlycate scores (normalized to the scale from 0 to 1) and the bootstrap-boosted frequencies of their occurrence in the examined groups of subjects, calculated with the use of the resampling adjusted for the sample size of the overall examined population ([*n* = 22 + 27 = 49]).

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
