# Peer review of "Fibrinogen Glycation and Presence of Glucose Impair Fibrin Polymerization—An In Vitro Study of Isolated Fibrinogen and Plasma from Patients with Diabetes Mellitus"

_biomolecules, 2020, doi:10.3390/biom10060877_

Round 1
Reviewer 1 Report
In the present MS, Authors Boguslawa Luzak investigated the impact of glucose and fibrinogen glycation on fibrin polymerization.
This is an interesting study using valuable techniques however many points hamper its publication in its present form.
Introduction
Lane 45 to 62, police seems smaller than in the rest of the section.
In this section, I think Authors should better develop on glycation phenomenon in term of reversible initial reactions, Amadori rearrangements …AGE formation. In addition emphasis can be made on initial glycation phenomenon and glycoxidative phenomena the latter involving oxidative stress generation.
Lane 59, sentence starting with “As lysine is involved in the cross-linking…” is too long and should be simplified.
Table 1. Age/sexe of participants should be added in the table.
Page 4 line 114, in the section 2.3. I understand fibrinogen was incubated in vitro in the absence or presence of 30 mM glucose during 4 days. Did they Author perform a control constituted by fibrinogen incubated with 5 mM glucose which corresponds to the physiologic glucose concentration?
Page 4 line 150 2.6. Fibrinogen purification. I understand Authors applied method from Dietrich et al. with some modifications to purify fibrinogen. How come no ammonium or polyethylene glycol precipitation steps are present in the protocol used by the Authors? By reading the protocol used in the MS I think steps seem missing to reach pure fibrinogen purification.
Page 4 line 158 Authors analysed fibrinogen purity on SDS Page gels. Data from gel image could be included in a supplementary data document.
Page 4 line 165. Please note the specificity in the fibrinogen glycation measurement with colorimetric NBT assay is based on the protein purity in the preparation.
Page 4 line 175, Authors measured carboxymethyllysine (CML) level in fibrinogen using western blot. I think example of blot image could be included in a supplementary data document.
Page 6 line 248 Authors wrote “antihyperglycsemic”
Result section:
Page 10 line 327. 3.4. Authors did not evidence any significant variation in fibrinogen glycation extend. Could data concerning CML level in total plasma be provided?
For the structural study of fibrinogen glycation, Authors provided information (figure 4 to 9) based on a software prediction. As LC MS/MS experiments were conducted, Authors could give important data concerning peptide glycation as changes in mass/charge by a more direct analysis of data obtained from spectrometer.
Page 19 line 535 Authors wrote “we conclude that the LCMS/MS technique does not allow accurate quantitative comparison of the degree of glycation between groups”. Such a postulate is not correct as many groups attested by high ranked publications have successively used MS/MS methodology to compare glycation degree between groups.
Same remark line 536, “the process of glycation is non-specific.” Such statement should not be done. Authors may write “in our experimental conditions…”
Page 19 Authors wrote “Our present findings identified an elevated glycation level (by about 30%) in diabetic fibrinogen or in fibrinogen incubated with glucose in vitro using the modified NBT assay. Actually results are not significant.
Line 437, Authors wrote about glycosylation, did they mean glycation?
English style may be improved.
Author Response
In the present MS, Authors Boguslawa Luzak investigated the impact of glucose and fibrinogen glycation on fibrin polymerization.
This is an interesting study using valuable techniques however many points hamper its publication in its present form.
- Introduction. Lane 45 to 62, police seems smaller than in the rest of the section. In this section, I think Authors should better develop on glycation phenomenon in term of reversible initial reactions, Amadori rearrangements …AGE formation. In addition emphasis can be made on initial glycation phenomenon and glycoxidative phenomena the latter involving oxidative stress generation. Lane 59, sentence starting with “As lysine is involved in the cross-linking…” is too long and should be simplified.
RE: According to Reviewer’s suggestion the Introduction extract was changed in the amended version of manuscript.
- Table 1. Age/sex of participants should be added in the table.
RE: According to Reviewer’s suggestion two values of age and the proportions of sex were added. Simultaneously, the information about age/sex were deleted from Material and Methods section, from point 2.2. Study population and blood collection (lines 89-91).
- Page 4 line 114, in the section 2.3. I understand fibrinogen was incubated in vitro in the absence or presence of 30 mM glucose during 4 days. Did they Author perform a control constituted by fibrinogen incubated with 5 mM glucose which corresponds to the physiologic glucose concentration?
RE: For the in vitro experiments a commercial preparation of human fibrinogen purchased from Calbiochem (Darmstadt, Germany) was used. According to the manufacturer’s information this fibrinogen is purified from plasma and not synthesized (this is not a recombinant protein). In our experimental protocol fibrinogen control samples were not additionally treated with 5 mM glucose, simultaneously we had in mind that the fibrinogen could be glycated under physiological condition like those occurring in blood from healthy volunteers before fibrinogen isolation.
- Page 4 line 150 2.6. Fibrinogen purification. I understand Authors applied method from Dietrich et al. with some modifications to purify fibrinogen. How come no ammonium or polyethylene glycol precipitation steps are present in the protocol used by the Authors? By reading the protocol used in the MS I think steps seem missing to reach pure fibrinogen purification.
RE: The experimental protocol for fibrinogen purification from human plasma did not incorporate ammonium or polyethylene glycol precipitation. Instead, we used fibrinogen precipitation with cold ethanol. In a scientific literature a few methods for fibrinogen purification from plasma were described, i.e. cryoprecipitation, ethanol precipitation, ammonium sulfate [(NH4)2SO4)] precipitation, and polyethylene glycol precipitation. According to Dietrich et al. publication (Tissue Eng Part C Methods 2013), ethanol precipitation is a simple and effective method for the isolation of fibrinogen with maximum fibrinogen yields of 80% of total plasma fibrinogen concentration. After fibrinogen isolation we tested its purity on SDS Page gels (the representative image is presented in supplementary materials) and we also verified the ability of isolated fibrinogens to clot; needless to say, these properties were kept.
- Page 4 line 158 Authors analysed fibrinogen purity on SDS Page gels. Data from gel image could be included in a supplementary data document.
Page 4 line 175, Authors measured carboxymethyllysine (CML) level in fibrinogen using western blot. I think example of blot image could be included in a supplementary data document.
RE: According to Reviewer’s suggestion the representative gel image with the samples of isolated fibrinogen and the Western blot image showing a CML-modified fibrinogen have been included in a supplementary data document (Figure S1).
- Page 4 line 165. Please note the specificity in the fibrinogen glycation measurement with colorimetric NBT assay is based on the protein purity in the preparation.
RE: The reviewer’s suggestion is very relevant, the contamination of the isolated fibrinogen samples with plasma proteins, such as albumin could increase the NBT-signal of protein glycation. However, to improve the specificity of measurements we did some experimental tricks. In our protocol we verified the purity of fibrinogen precipitation on SDS PAGE electrophoresis. Additionally we quantified the concentration of purified fibrinogen spectrophotometrically (280 nm) with a coefficient of extinction 1.55 for a fibrinogen. According to Gugliucci et al. to improve measurements of glycated fibrinogen with nitroblue tetrazolium the heating (56oC) and the presence of a detergent (Zwittergent) are helpful and we included these modifications into the assay.
- Page 6 line 248 Authors wrote “antihyperglycsemic”
RE: Thank you for the suggestion. The mistake was corrected in the new version of manuscript (word “antihyperglycsemic” was replaced by “ antihyperglycaemic”).
Result section:
- Page 10 line 327. 3.4. Authors did not evidence any significant variation in fibrinogen glycation extend. Could data concerning CML level in total plasma be provided?
RE: We performed a Western blot analysis of CML-modified proteins in plasma of diabetic patients and control subjects to compare amount of CML-modified proteins between the groups. However, we were not able to determine an absolute amount of CML-modified proteins in the plasma samples because a Western blot is a semiquantitative method for detecting electrophoretically separated proteins. Given the relative amount of CML-modified proteins calculated in the course of experiments, we did not demonstrate significant differences between diabetic and control plasma, and we decided not to show these results. In this report, which concerns the fibrinogen per se, we showed the results obtained only for this protein.
- For the structural study of fibrinogen glycation, Authors provided information (figure 4 to 9) based on a software prediction. As LC MS/MS experiments were conducted, Authors could give important data concerning peptide glycation as changes in mass/charge by a more direct analysis of data obtained from spectrometer.
RE: According to Reviewer’s suggestion some data concerning peptide glycation in LC MS/MS analysis were included into Materials and Methods paragraph in a new version of manuscript.
- Page 19 line 535 Authors wrote “we conclude that the LCMS/MS technique does not allow accurate quantitative comparison of the degree of glycation between groups”. Such a postulate is not correct as many groups attested by high ranked publications have successively used MS/MS methodology to compare glycation degree between groups.
Same remark line 536, “the process of glycation is non-specific.” Such statement should not be done. Authors may write “in our experimental conditions…”
RE: According to Reviewer’s suggestion the statement was changed (in the new version of manuscript is: In our experimental conditions and upon analyzing of several samples from patients with and without diabetes, we have noted that the LCMS/MS technique does not allow accurate quantitative comparison of the degrees of glycation between groups, as far as the process of glycation is non-specific.
- Page 19 Authors wrote “Our present findings identified an elevated glycation level (by about 30%) in diabetic fibrinogen or in fibrinogen incubated with glucose in vitro using the modified NBT assay. Actually results are not significant.
RE: In our study the level of fibrinogen glycation measured with NBT assay was more profound for in the vitro experiment, and under these conditions the difference in fructosamine amount between non-treated and glucose-treated samples was statistically significant (the paragraph 3.3). Unfortunately, because of a big variability of data from both control and diabetic groups, the difference in fibrinogen glycation was not statistically significant, although on average it reaches above 20% difference.
- Line 437, Authors wrote about glycosylation, did they mean glycation?
RE: The unfortunately used word “glycosylation” was replaced by “glycation” in the new version of manuscript. The manuscript concerned on glycation, sometimes called non-enzymatic glycosylation.
- English style may be improved.
RE: Manuscript has been edited by native speaker.
Reviewer 2 Report
In their manuscript, Luzak et al., analyzed the effects of fibrinogen glycation on fibrin polymerization in plasma of patients with T2DM diabtes mellitus. The authors show that the polymerization velocity is lower in the plasma of these patients compared to that of control subjects.
The authors explain the results that glycation of fibrinogen may influence clotting ability of fibrinogen during its polymerization. The authors discuss that such an impairement may influence diagnostic results. The study is interesting and methods and results are well presented.
I have following concerns:
- It would be important that the authors present scanning electron of fibrin clots from control subjects and T2DM, plasma incubated in presence and absence of glycose, and thrombin.
- It would be also important to discuss role of platelets in diabetes. Platelets contain fibrinogen in their granules. This fibrinogen released upon activation in patients can be also glycated.
- What could be effect of glycose on clot retraction ? in vitro clot retraction assay
Author Response
In their manuscript, Luzak et al., analyzed the effects of fibrinogen glycation on fibrin polymerization in plasma of patients with T2DM diabetes mellitus. The authors show that the polymerization velocity is lower in the plasma of these patients compared to that of control subjects.
The authors explain the results that glycation of fibrinogen may influence clotting ability of fibrinogen during its polymerization. The authors discuss that such an impairment may influence diagnostic results. The study is interesting and methods and results are well presented.
I have following concerns:
- It would be important that the authors present scanning electron of fibrin clots from control subjects and T2DM, plasma incubated in presence and absence of glycose, and thrombin.
RE: The reviewer’s suggestion is very relevant. The proposed experiments could possibly add new data to our way of reasoning. However, due to the epidemiological situation in our country, the core facility which is capable to perform SEM is locked down until further notice. Therefore, we have no technical ability to perform the suggested experiments.
2. It would be also important to discuss role of platelets in diabetes. Platelets contain fibrinogen in their granules. This fibrinogen released upon activation in patients can be also glycated.
RE: As the reviewer mentioned, platelets play very important role in cardiovascular complications in diabetes. Probably some amount of platelet-released fibrinogen also takes part in a thrombus formation, but there is little experimental data on this aspect. Furthermore it is not clear whether platelet fibrinogen is glycated under hyperglycaemic conditions, as far as this problem has not been reported in scientific literature yet. In our study we have focused on plasma fibrinogen glycation and its role in clot formation having in mind that the ratio of platelet to plasma fibrinogen in whole blood was determined to be 1:30 (Lopaciuk et al., Thromb Res 1976, 8(4): 453-65). We agree with the Reviewer that platelet fibrinogen and its possible glycation in hyperglycemia may influence thrombus formation in physiological conditions. We have added this consideration in the amended version of the manuscript.
3. What could be effect of glycose on clot retraction ? in vitro clot retraction assay
RE: According to Reviewer’s suggestion, the experiment concerning clot retraction in the presence of 5 or 30 mM was done. The results have shown the decreased percent of clot retraction and increased the weight of clot in the samples with 30 mM glucose compared to control (without glucose), but the differences were under statistical significance. The relevant data were described in Table S1 in supplementary and comment is included in the Results paragraph (3.3) in the amended version of manuscript.
Reviewer 3 Report
The authors present an in-vitro study on the effects of chronic hyperglycemia on biochemical properties of fibrinogen. This work is highly relevant, as alterations in the coagulatory systems in T2DM patients are well-known, but poorly understood. It is especially surprising, that the authors report an impairment of fibrinogen action, which somehow contradicts the common finding of hypercoagulability in T2DM.
Introduction is sufficiently presenting the scientific background for this paper.
The methods section is mainly conclusive. Given the skewed and incongruent sex ratio in both healthy (women>>men) and T2DM (men>>women) subjects, statistical adjustment for or stratification by sex is warranted.
For the results, ll. 240-242 are somehow misunderstandable, as the 27 T2DM subjects are characterised as either long-term (> 5 years, n=9) or short-term (<1 year, n=9: "the other nine") diabetics, but this does not sum up to 27 patients. Similarly, the three treatment groups (insulin +/- other OADs, metformin or sulfonylureas) sum up to 28 subjects, not 27. Blood pressure levels are implausible, given that all values reported are multiples of 10. For transaminases, consistent use of one abbrevation (AST or ASAT or ASPAT; same for ALT) is recommended.
Any relation between continuous glycemic variables (HbA1c, fasting glucose, artificially set glucose levels) and coagulatory outcomes (TT, PTT, aPTT, fibrinogen levels, Vmax...) should be checked for non-linear associations by creating scatter plots in order to outrule an "satiation phenomenon" when reaching higher glucose levels and to make sure that there is no potentially higher actual cut-off for differences in coagulation parameters (other than the conventional discriminant between healthy and T2DM).
A review of the discussions section can be finalised after minor revision, addressing the few aforementioned aspects for clarification of potentially missed associations.
Author Response
The authors present an in-vitro study on the effects of chronic hyperglycemia on biochemical properties of fibrinogen. This work is highly relevant, as alterations in the coagulatory systems in T2DM patients are well-known, but poorly understood. It is especially surprising, that the authors report an impairment of fibrinogen action, which somehow contradicts the common finding of hypercoagulability in T2DM.
Introduction is sufficiently presenting the scientific background for this paper.
The methods section is mainly conclusive. Given the skewed and incongruent sex ratio in both healthy (women>>men) and T2DM (men>>women) subjects, statistical adjustment for or stratification by sex is warranted.
For the results, ll. 240-242 are somehow misunderstandable, as the 27 T2DM subjects are characterised as either long-term (> 5 years, n=9) or short-term (<1 year, n=9: "the other nine") diabetics, but this does not sum up to 27 patients. Similarly, the three treatment groups (insulin +/- other OADs, metformin or sulfonylureas) sum up to 28 subjects, not 27. Blood pressure levels are implausible, given that all values reported are multiples of 10.. For transaminases, consistent use of one abbrevation (AST or ASAT or ASPAT; same for ALT) is recommended.
RE: The reviewer’s suggestion is very relevant. The data were corrected in the amended version of manuscript. Also, the abbreviations for transaminases were changed.
The blood pressure measurements were carried out in accordance with the applicable standards of the European Cardiac Society (2018 ESC/ESH Clinical Practice Guidelines for the Management of Arterial Hypertension; Eur Heart J 2018 Sep 1;39(33): 3021-3104). In our study, a certified digital blood pressure monitor was used with a pitch every 2 mmHg, not an electronic one. The obtained values were rounded to 5 or 10 mmHg.
Any relation between continuous glycemic variables (HbA1c, fasting glucose, artificially set glucose level and coagulatory outcomes (TT, PTT, aPTT, fibrinogen levels, Vmax...) should be checked for non-linear associations by creating scatter plots in order to outrule an "satiation phenomenon" when reaching higher glucose levels and to make sure that there is no potentially higher actual cut-off for differences in coagulation parameters (other than the conventional discriminant between healthy and T2DM).
RE: Based on the scatter plots we reasoned on the occurring data asymmetry. For the assessing of the associations we used the rank procedures (rank Spearman, Kendall or gamma) and that ruled out the “problem” of data asymmetry. The additional validation for the obtaining of significant associations not merely by a pure chance was gathered with the use of resampling techniques. In general, the variables departing from normal distributions were transformed (Box-Cox transformation or others) and the linear model analyses were performed on the transformed data. For the reason of relatively low sample sizes, we additionally used the approach of bootstrap-boosted calculation procedures in order to minimize the possibility that any associations were demonstrated by a pure chance. When checking for the statistical differences between the groups, we generally used the bootstrap-boosted ANCOVA with the adjustment for sex and age. Moreover, in the cases where the groups demonstrated significant differences even upon the adjustment, we performed the bootstrap-boosted ANCOVA analysis with the sample size adjustment for the equalized proportions of men and women (Table 1, Table 2). The associations between coagulation parameter (APTT, PT, TT) were described in the Results section (pp.8) (there were no significant associations). Also, in the Table 3 we have shown the association between slope and glycaemia or fructosamine concentration.
A review of the discussions section can be finalised after minor revision, addressing the few aforementioned aspects for clarification of potentially missed associations.
RE: Of course, the Discussion section was modified upon revising the document according to the Reviewers’ comments.
Round 2
Reviewer 1 Report
In their revised version of the MS, Authors correctly addressed points I previously raised.